



# Does the pace of carbon emissions matter in an atmospheric general circulation model?

Anja Katzenberger[1,2] and Anders Levermann[1,2]

[1]Potsdam Institute for Climate Impact Research, Potsdam, Germany
[2]Institute for Physics and Astronomy, Potsdam University, Potsdam, Germany

**Correspondence:** Anja Katzenberger (anja.katzenberger@pik-potsdam.de)

**Abstract.** The concentration of carbon dioxide in the atmosphere changes our climate and its variability. It impacts all parts of nature and society. Consequently, there is an ongoing societal discussion about speeding up the transition to net-zero carbon emissions. The faster emissions are reduced, the less carbon dioxide will accumulate in the atmosphere where it largely remains influencing climate for hundreds of years. What has not yet been broadly studied, is the question whether the rate of the
emissions themselves in addition to the resulting concentration has significant impact on climate and weather. To address this question, we run simulations with the Geophysical Fluid Dynamics Laboratory Atmospheric Model version 2 (GFDL-AM2), with different $CO_2$ forcing rates. In order to investigate mainly the atmospheric role, the oceanic boundary condition is supplied by a slab ocean. We find that for the the same warming level (2°C) but different warming rates (e.g. annual rates of 1% $CO_2$ increase compared to 4%), the differences in the annual average temperature and precipitation and day-to-day variability
patterns are of the same order of magnitude between different simulations with the same rate and between different simulations with different rates. Thus, we find that without a significant influence of ocean circulation changes, the fast mixing times within the atmosphere and thereby the lack of an atmospheric memory, inhibits a significant role of the rate of $CO_2$ emissions for weather variability. This result is not unexpected, but needed confirmation. In summary, the atmospheric dynamics alone do not allow for an influence of the rate of carbon emissions on the annual average and day-to-day variability in temperature and
precipitation.

## 1 Introduction

Earth's climate is changing due to the burning of oil, gas and coal. The underlying key relationship between the amount of atmospheric carbon dioxide and the global mean temperature is well understood (Masson-Delmotte et al., 2021). This relationship was initially proposed in the 19th century (Arrhenius, 1896), is understood within the realm of quantum and
statistical physics and has since been recorded by a multitude of measurement stations and satellites within the 20th century NASA (2024). The fundamental physical understanding of the phenomenon is implemented in numerical atmospheric general circulation models and projected into the 21st century and beyond for different emission scenarios (Masson-Delmotte et al., 2021). In addition to these impacts of $CO_2$ on mean climatic conditions, understanding climate variability may be even more relevant (Katz and Brown, 1992; Schär et al., 2004) as the increased frequency and intensity of extremes, coupled with reduced





predictability, pose great challenges for adaptation and lead to significantly higher damages. In this context, changes in climate variability have been found to reduce economic growth (Kotz et al., 2021b), be relevant for human health (Shi et al., 2015; Zanobetti et al., 2012), behaviour (Stechemesser et al., 2022) and agricultural productivity (Wheeler et al., 2000; Rowhani et al., 2011; Ceglar et al., 2016; Asseng et al., 2011; Gallé and Katzenberger, 2024).

The year-to-year variability of the global mean temperature (GMT) under climate change has been discussed from different
perspectives and region-dependent trends have been identified: In Europe, the interannual temperature variability is projected to increase, particularly during summer months (Guo et al., 2021; Fischer et al., 2012; Schär et al., 2004; Kitoh and Mukano, 2009). Also in India and East Asia interannual precipitation variability is robustly projected to increase in CMIP6(Katzenberger et al., 2021; Katzenberger and Levermann, 2024). By contrast, a decreasing trend has been identified e.g. in the Northern hemisphere high latitudes in winter (Kitoh and Mukano, 2009). Despite these identified regional patterns, no final conclusions have
yet been reached on the global level (Lenderink et al., 2007; Fischer and Schär, 2009; Kitoh and Mukano, 2009; Huntingford et al., 2013).

On the day-to-day scale, reanalysis data of historical observations and climate models agree that the variability over the past decades has been increasing at low to mid latitudes (Kotz et al., 2021a; Kitoh and Mukano, 2009) and decreasing at northern mid to high latitudes (Karl et al., 1995; Michaels et al., 1998; Screen, 2014; Kotz et al., 2021a; Ylhäisi and Räisänen, 2014).
Under unabated greenhouse gas emissions, this trend is projected to continue by up to an additional 100% at low latitudes and by 40% at northern high latitudes until the end of the 21st century (Kotz et al., 2021a). Analysis based on the multi-model ensemble from the Coupled model Intercomparison project Phase 6 confirm that these changes can be attributed to enhanced greenhouse gas forcing (Kotz et al., 2021a).

While the change in the mean and variability for different greenhouse gas emission scenarios has been analysed, the question
remains whether the pace of these emissions plays a special role for the annual mean climate and its short-term, i.e. day-to-day, variability. In this study, we simulate different rates of $CO_2$ increase using the Geophysical Fluid Dynamics Laboratory Atmospheric Model version 2 (GFDL-AM2). The historic increasing rate was approx. 0.5% ppm/year (1978-2023) (Lan and Thoning, 2023) and is projected to reach up to 1.3% under SSP5-8.5 scenario (until 2100). Therefore, we use a 1% scenario, that we complement by a 2%, 3% and 4% scenario. We then compare the simulations concerning their 10-years average and day-
to-day variability. In section 2 we provide an overview of the methods applied including the model setup (2.1), the simulations performed (2.2) as well as the procedure to remove trend and seasonality for the variability analyses (2.3). In subsection 3.1. and 3.2 we analyse the differences in the mean temperature and precipitation for the 10-years period at a global warming level of 2°C. Subsection 3.2 presents the results for temperature variability and subsection 3.3 focuses on precipitation variability before concluding in Section 4.





## 2 Methods

### 2.1 The model setup

We use the atmospheric component GFDL-AM2 of the general circulation model that is developed at the Geophysical Fluid Dynamics Laboratory (GFDL) of the National Oceanic and Atmospheric Administration (NOAA). The focus in the development process was on creating a model with realistic representation of the dynamic, thermodynamic and radiative components of the climate system (Anderson et al., 2004). The horizontal grid resolution is 2° latitude x 2.5° longitude. The vertical grid consists of a hybrid coordinate grid with 24 vertical levels ranging from about 30 m above the surface to the top level at about 3 hPa. Advective and physics time steps are 10 minutes and 0.5 hours (Anderson et al., 2004), for atmospheric radiation 3 hours time steps are used in order to include a diurnal cycle (Delworth et al., 2006).

Since in this study we focus on atmospheric processes, we use a highly simplified "slab-type" ocean model (Knutson, 2003; GFDL Team, 2019) rather than a computationally intensive ocean general circulation model. The horizontal grid points in this module represent slabs of water with uniform depth (here 200 m) and salinity (33.33 parts per thousand). The sea surface temperature at each grid point is determined by the heat exchange across the air-sea and ocean-sea ice interfaces. Between the grid cells, there is no communication (thus no ocean currents, temperature advection, diffusion or convection). Corresponding to the use of the slab ocean model, the sea-ice model (Winton, 2000) is also run in "slab mode".

To represent land dynamics, we use the module LaD (Milly and Shmakin, 2002) that is originated in the early model development of (Manabe, 1969), who simulates the continents as boxes with limited water storage.

### 2.2 Simulations with the GFDL-AM2

After equilibration, we run the GFDL-AM2 model in the described setup with constant atmospheric $CO_2$ of 280ppm for 20 years to obtain a period with equilibrated conditions. Starting from 280ppm, we perform simulations with different rates including 1%, 2%, 3% and 4% increase per year for 60 years each (Fig. 1). Since the 4% simulation reaches very high and unrealistic values that lead to numerical instabilities, these simulations are limited to 50 years. We use daily 2m surface air temperature and precipitation output. In order to check for robustness, we also repeat the 1% and 4% four times with varying initial condition (starting the increase one year later each), to receive an ensemble of five simulations (denoted ens1, ens2, ens3, ens4 and ens5). It has to be noted that the sea ice cover in these simulations reaches too far southwards on the NH, and too northwards on the SH. As we focus on differences compared to the initial state, this issue is less relevant for our analysis.

### 2.3 Removing trend and seasonality

For each of the four 50-years model output data with varying forcing rates, we identify the 10-years window around the time step when an increase of 2°C GMT is reached. For the mean analysis, for each grid cell, we calculate the annual time series of temperature and precipitation and subtract the annual mean of the first year to obtain the difference compared to the beginning of the simulations. For the identified 10-years window, we then calculate the mean for each grid cell. The weighted spatial





**Figure 1. Overview of the simulations by the GFDL-AM2 model with different rates of CO₂ increase** Left panels: CO₂ forcing starting from preindustrial reference of 280ppm. Middle panels: Daily global mean temperatures (GMT; orange) with their annual means (red). The vertical dashed lines marks the first time step that exceeds a temperature increase of 2°. The gray background marks the resulting 10-years period that is used for the analysis. Right panels: As the middle panels but for precipitation.





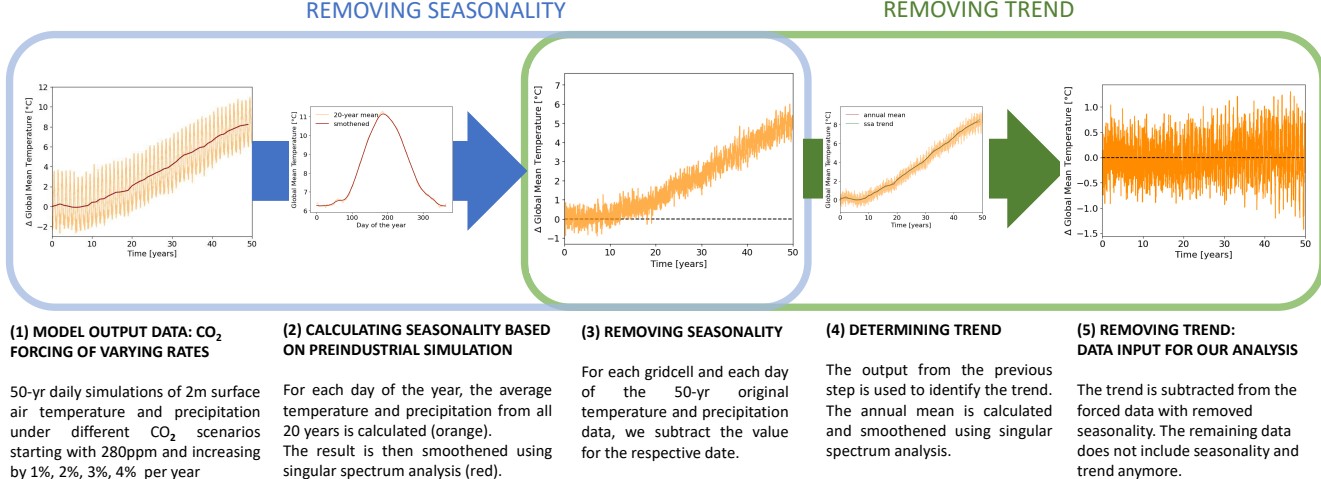

**Figure 2. Conceptual summary of the methods applied for calculating variability.** Applied procedure to remove the seasonality and trend from the four simulations with different $CO_2$ forcing.

average across the globe is around 2°C with small deviations. From each grid cell, we subtract the global spatial average in order to obtain data centered around 0.

For the variability analysis, we remove trend and seasonality following the procedure summarized in Fig. 2. The 20-yrs period of equilibrated conditions serve to calculate the seasonality of temperature and precipitation for each grid cell: For each

date and each grid cell, we average over all 20 years and smoothen the results by singular spectrum analysis (Golyandina and Zhigljavsky, 2013). For each timestep and each gridcell in the 50-yrs simulations, we then subtract the value for the respective date to remove seasonality. As a next step we determine the trend of the remaining timeseries by calculating the annual means and smoothen it by using singular spectrum analysis. We then subtract the trend and use the identified 10-years window as a basis for our analysis.

## 3 Results

### 3.1 Differences in Mean Temperature Pattern

In the simulations with higher rates of $CO_2$ increase, the global mean temperature rises faster compared to the other simulations, (Fig. 1). In order to analyse the annual mean climate state and its day-to-day variability, we use the 10-year window around the year when a 2°C increase of GMT is reached. In the simulation with a $CO_2$ increase of 1% per year (we refer to this simulation

as '1% simulation'), this increase is reached after 33-40 years, in the different ensemble member (ens1: 42 years, ens2: 40 years, ens3: 38 years, ens4: 40 years, ens5: 33 years). In the one 2% simulation it takes 29 years, 23 years in the 3% simulation and 20-22 years in the 4% simulation (ens1: 20 years, ens2: 21 years, ens3: 20 years, ens4: 22 years, ens5: 20 years).





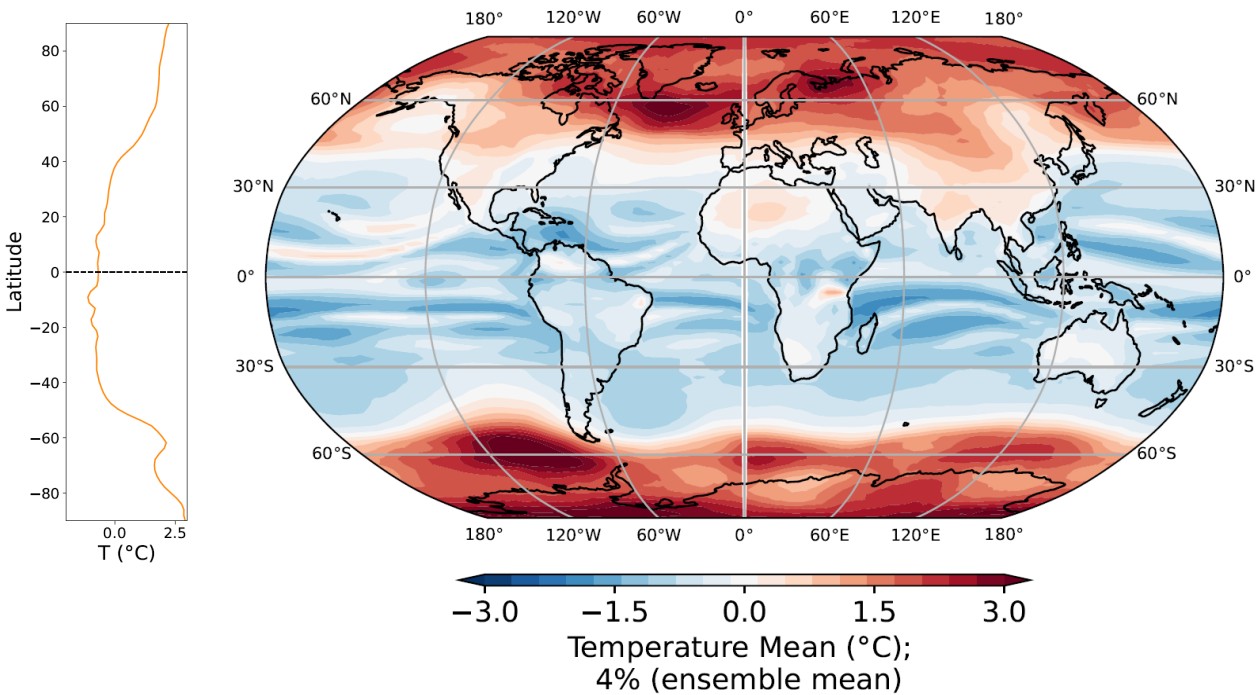

**Figure 3. Global (right) and meridional (left) distribution of temperature (10-years average) for the 4% ensemble.** The mean from all five ensemble members is shown. The 1% ensemble is shown in Fig. A1 and individual ensemble members in Fig. A3 and A2.

The 10-year temperature averages around the global warming level of 2°C in all simulations are dominated by a strong warming signal in higher latitudes compared to low and mid latitudes, as exemplary shown for the mean of the 4% ensemble in Fig. 3 and for the 1% ensemble in Fig. A1. This feature is also present in the individual simulations, with varying regional intensities across the high latitudes, see Fig. A2 and A3. Another omnipresent characteristic are the stripes of comparatively colder temperature in the tropics, particularly over the ocean. These stripes are present in the individual simulations, as well as the ensemble mean and may indicate the location of the tropical rainfall band. Note that the role of the ocean is only that of a heat bath. It can absorb and release energy but there is no oceanic circulation only vertical diffusive exchange. In order to show the remaining patterns without the overlying polar amplification effect, we subtract the zonal averages from the spatial distribution (Fig. 3) which reveals the tendency of the land to warm faster than the ocean, see Fig. A4 and A5.

Comparing the two ensembles with the forcing rates of 1% and 4% shows differences of up to 2°C. The signal in the very high latitudes suggests that the polar amplification is stronger in the 4% simulations, however the region is limited to few grid cells in the polar regions (>75°), see Fig. 4 (upper panel). Even more importantly, combining different individual ensemble members shows that the spatial patterns are not at all consistent across simulations within the same ensemble (Fig. A6). We find that the differences between the mean states of the 4% and 1% ensemble are of the same magnitude as the one within the



**Figure 4. Difference in global temperature distribution between the 4% and 1% simulations.** Upper panel: Difference between ensemble means; Lower left panel: Difference between one member of the 4% ensemble, and one member of the 1% ensemble; Lower right panel: Difference between two members from the same ensemble. Further combinations are shown in Fig. A6 and A7.





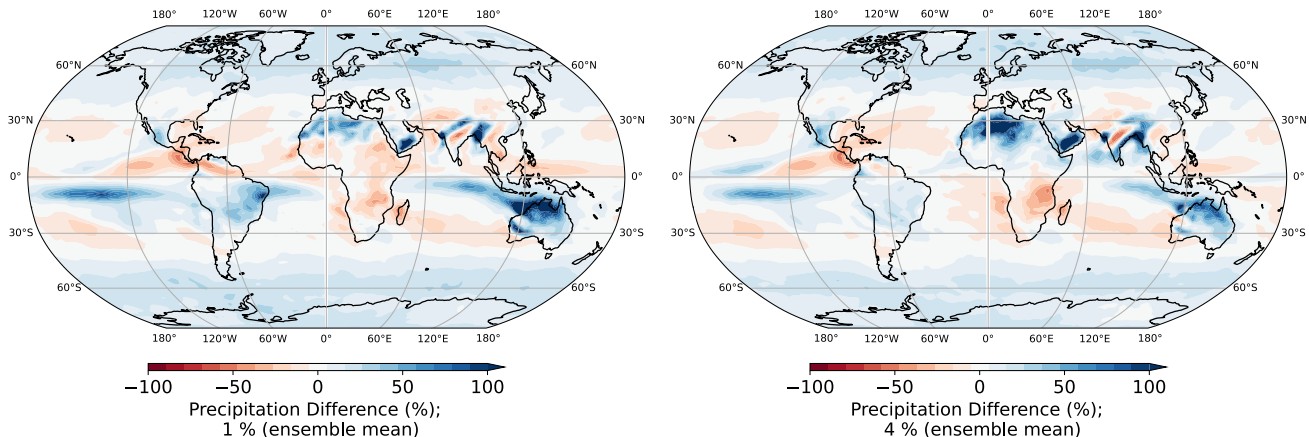

**Figure 5. Relative changes in global precipitation at a global warming level of 2°C compared to preindustral equilibrium.** The ensemble means for 1% ensemble (left) and 4% ensemble (right) are shown.

same ensemble (Fig. A6 and A7). This indicates that the differences between the temperature mean states are not significant as they do not exceed the internal variability.

It is important to note that the 10-year average patterns are strongly different within each ensemble of the same warming rate

and have the same order of difference between states of different warming rates. This emphasizes the known fact that there is internal variability in the climate state even on time scales of a decade. This is the reason why long-term trends in the climate state can only be detected for periods of 20-30 years as already stated in the first assessment report of the Intergovernmental Panel on Climate Change (IPCC, 1990).

### 3.2   Differences in Mean Precipitation Pattern

The relative change in global precipitation patterns at a global warming level of 2°C (10-years average) compared to preindustrial levels of 280ppm (20-years average) shows varying trends for different regions (Fig. 5). The 4% and 1% ensemble mean show very similar changing patterns suggesting that there are no major differences between the simulations with different $CO_2$ forcing rates. Due to the high rates of precipitation in the tropics the absolute difference shows a dominating signal in the ITCZ. In the 4% ensemble mean, the precipitation intensifies in 0-10°N by more than 2 mm/day while it decreases in

0-10°S by a similar amount indicating a northwards shift of the ITCZ (Fig. 6 (left)). This northward shift is present in all five members of the 4% ensemble (Fig. A9). The 1% ensemble mean represents a more complex picture (Fig. 6 (right)) across its five ensemble members (Fig. A8). One ensemble member even shows a southward shift of the ITCZ (ens1), others show partly the same behaviour as in the 4% simulations or other. This difference between the 4% and 1% simulations is important to note, but is most likely due to a statistical sampling problem than due to the different warming rates. Since the precipitation pattern is strongly linked with the temperature pattern which does not show robust differences, these findings regarding precipitation

are very likely not significant. Further indication that the differences are not originated in different warming rates, is that the





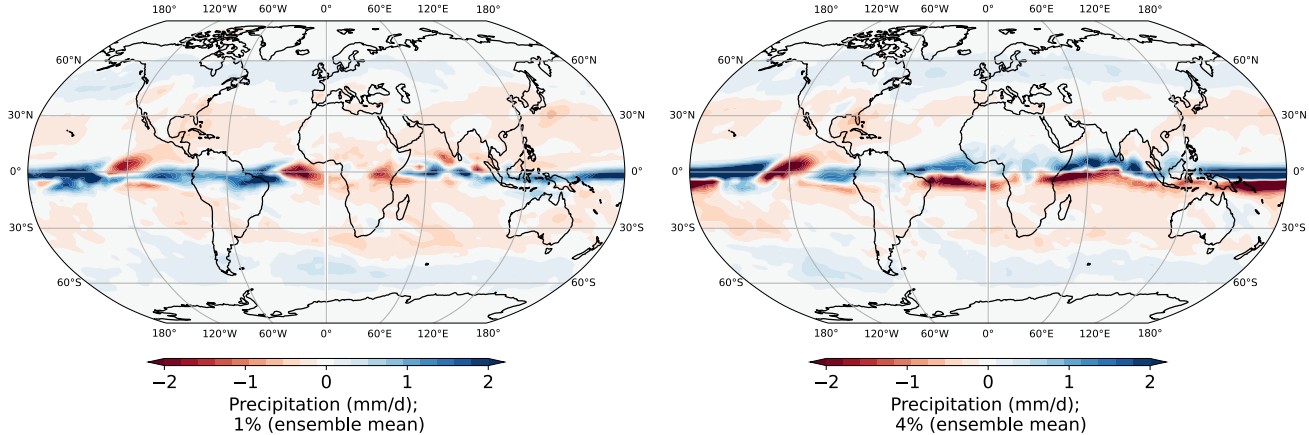

**Figure 6. Global distribution of precipitation changes (10-years average) compared to the preindustrial equilibrium.** The mean from all five ensemble members is shown for the 1% (left) and 4% ensemble (right). The individual ensemble members are shown in Fig. A8 and A9.

differences between two members of the 4% and 1% ensemble are of the same order of magnitude as the differences within one ensemble.

### 3.3 Comparison of temperature variability pattern

For this 10-yrs time period with comparable mean states for the same warming level, we analyse the day-to-day temperature variability (Fig. A10 exemplary for the 1% ensemble). The temperature variability is higher over land compared to the ocean as a result of different thermal capacities of land and ocean. It increases from tropical regions towards the poles where it reaches a standard deviation of approx. 10°C. Focusing on the NH summer (June to August, JJA) or the NH winter (December to February, DJF), it becomes clear that the regions of highest variability are associated with the change in sea ice cover.

Sea ice changes result in strong albedo changes and thus temperature changes, while the open ocean has higher heat capacity, is thus buffering temperature changes and is consequently associated with less near surface atmospheric temperature variability.

Comparing individual members of the 4% with the 1% ensemble reveals differences in the relative standard deviation exceeding 25%. Fig. 7 (left) illustrates exemplary the difference between the first members from both ensembles. In this specific pair

of simulations, for example, the 4% simulation has higher variability in high northern latitudes in the annual average, while the high southern latitudes exhibit higher variability in the 1% simulation. In the tropics, there are strong differences between the simulations with different forcing paces - depending on the specific region. However, when combining different members from the 1% and 4% ensemble, it becomes clear that the regional patterns are not consistent across ensemble members (Fig. A11). Also combinations involving the 3% and 2% simulations show varying regional patterns (Fig. A12). Addition-

ally, the comparison between two members from the same ensemble reveals that the difference between simulations with the




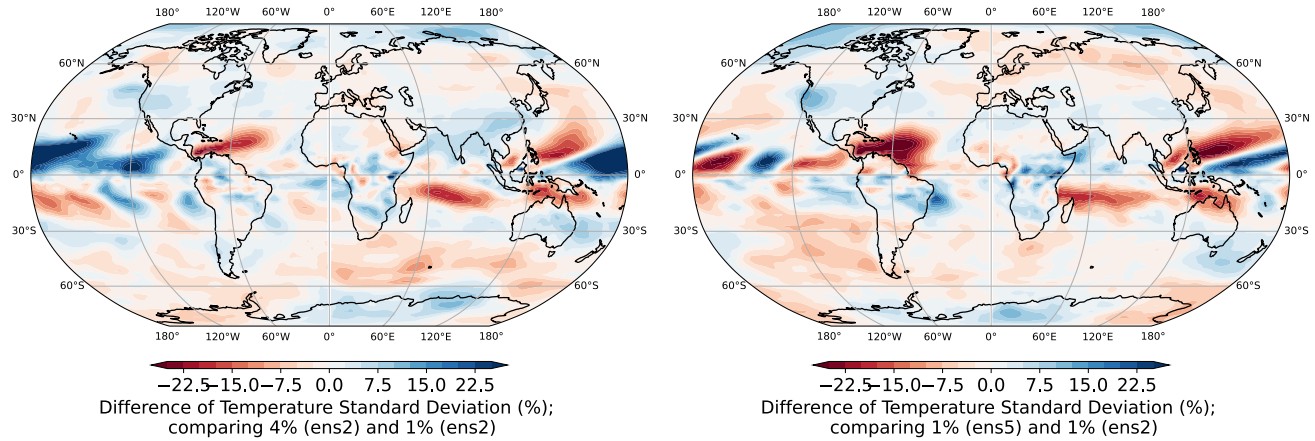

**Figure 7. Relative difference in the day-to-day temperature variability between 4% and 1% simulations for entire year (left), and two members of the 1% ensemble (right).** As the differences are of the same order of magnitude, it becomes clear that they are not significant. Further ensemble combinations are shown in Fig. A11.

same forcing pace are of the same magnitude as the difference between simulations with different forcing paces (Fig. 7 (right)).

This clearly indicates that the differences in variability between the simulations with different carbon emission rate does not arise from the different warming paces, but reflect the differences in the variability itself. In order to proof that the differences
observed between the different ensemble members of the 1% simulations and the 4% simulations are statistically not significant, would require a much larger ensemble than 5 members for each pace. This is due to the strong variation in the variability itself and is beyond the computational capacity that we can provide. We thus have to simply illustrate that the differences between simulations from different ensembles show the same magnitude of variability as differences between simulations of the same ensemble.

**3.4 Comparison of precipitation variability pattern**

The day-to-day variability in precipitation is strongest in the Intertropical Convergence Zone (ITCZ) (Fig. A13). The standard deviation there reaches up to 20 mm/day in the annual average. Increased variability also takes place in the mid latitudes where the storm tracks are located. The individual seasons show predominantly varying intensity along the latitudes, but also along longitudes: In JJA there is a higher variability in the equatorial Atlantic ocean, while in DJF the equatorial Indian ocean shows
increased variability.

As a consequence of the strong variability within the ITCZ, it is not surprising that also the differences between individual members of the 4% and 1% simulations shows the highest signals in this region. For example the day-to-day variability for the example of one specific pair of ensemble members between the 1% and 4% simulations shows that variability is higher for





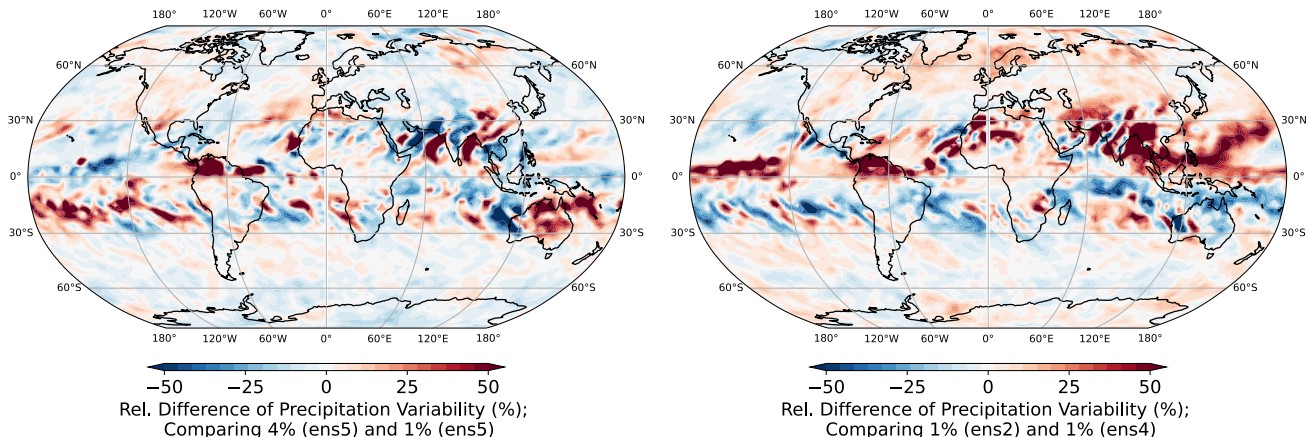

**Figure 8. Relative difference in day-to-day precipitation variability between 4% and 1% simulations for entire year (left), and two members of the 1% ensemble (right).** As the differences are of the same order of magnitude, it becomes clear that they are not significant. Further ensemble combinations are shown in Fig. A14.

the faster emission scenario in the NH tropics (even more than 50% in some regions), while the variability in the SH tropics is clearly higher in the simulations with slower rates of $CO_2$ increase (Fig. 8 left). Over the ITCZ Indian Ocean, these difference are enhanced from October to February, while over the equatorial Atlantic the anomalies are strongest during JJA. The fact that highest variability differences are found in the ITCZ associated with the Hadley circulation is also present when combining different members of the 4% and 1% ensemble (Fig. A14) or combining 3% and 1%, 2% and 1% or other combinations (Fig.

A15). The order of magnitude of these changes is the same between two simulations of the same ensemble (Fig. 8 right) and between different ensembles. Therefore, in conclusion, there are no significant differences in the precipitation variability pattern between the simulations with different forcing paces. As expected, such changes do not emerge merely from the atmospheric dynamics or thermodynamics due to its fast mixing time scales.

## 4   Discussion and Conclusion

In this study, we address the question whether the pace of $CO_2$ emissions has an impact on the climate averaged over a decade and day-to-day weather if the ocean circulation as a carrier of long-term memory is excluded. Our setup is designed to specifically focus on the role of the atmosphere simulated by the atmospheric circulation model GFDL-AM2. Therefore, the ocean is simulated merely as a boundary condition for the atmosphere without internal dynamics and is therefore represented by a simplified slab ocean that excludes oceanic circulations, especially meridional circulations and deep water formation.

Starting from an equilibrated state with an atmospheric $CO_2$ concentration of 280ppm, we run simulations with different forcing paces ranging from 1% to 4% to analyze the 10-years periods around the same warming level. After correcting for trend and seasonality, we calculate the day-to-day variability for each grid cell. Comparing simulations with different forcing paces





regarding decadal average and day-to-day variability reveals differences of the same order of magnitude as when comparing different simulations of the same ensemble. This does not proof but strongly suggests that there are no significant differences

between the simulations with different forcing paces for the decadal temperature and precipitation averages nor for their variability patterns, that emerge only from the atmospheric dynamics alone.

The design of our model configuration with the clear focus on the atmosphere is relevant to understand why there are no significant differences between the simulations with different rates of forcing. The simplified slab ocean strongly reduces the complexity of the ocean to a mere heat bath. Therefore, any differences in the variability must be originated in the atmo-

spheric capacity to memorize the different forcing history. However, as was expected and is strongly suggested by our results, the atmospheric mixing time scales are too fast to allow for such a long-term memory that would allow the atmosphere to 'remember' the last year. The ocean however is known for its very long mixing time scales and it is possible and likely that when using a full atmospheric-ocean circulation model, differences between the simulations with different forcing history will occur. For faster forcing paces, the GMT will warm the ocean faster at the surface leading to changes in the ocean circulation

including convection and deep water formation. Repeating these simulations including an oceanic gerneral circulation model is an important next step to take, but is outside of the scope of this study.

By working with an ensemble of five simulations for the 1% and 4% forcing pace, we get insights regarding the robustness of our results across different realizations within the GFDL-AM2 model. Such ensembles based on simulations with varying initial conditions are established tools to get insights concerning natural variability (Maher et al., 2021). It is noteworthy that

the spatial differences even of the decadal mean between different members of the same ensemble are significantly. This means that the atmospheric variability alone can yield very different regional distributions of temperature and precipitation for up to a decade.

Regarding the robustness of our results, it also has to be acknowledged, that we only use one model, namely the GFDL-AM2. It's known that each model has its own (or common) biases and that therefore multi-model ensembles often outperform

individual models with respect to the comparison with observations (Gleckler et al., 2008; Reichler and Kim, 2008; Pincus et al., 2008; Lambert and Boer, 2001; Palmer et al., 2005). However, simply averaging across such a multi-model ensemble in our context might also lead to lose potential signals that only occur in some of these models. Therefore, models results would need to be checked individually and compared carefully. However, it would certainly be very interesting and - given the relevance of the question- also an important opportunity to repeat this or a similar analysis with different general circulation

models.

While this study suggests that there are no differences in the mean and variability between simulations with different forcing paces, the implementation of a full ocean is likely to lead to the finding that a faster pace itself is also associated with different mean states and different variability. In any case it has to be noted that faster $CO_2$ emissions will result earlier in higher global mean temperatures, higher mean precipitation and associated higher variability. Thus, all adverse affects from an increase in

the global mean temperature will impact society and nature faster under faster warming. Adapting to such conditions with less time poses diverse challenges. It is likely that increasing the pace of warming will eventually mean that impacts will outpace





the adaptive capacities of ecosystems (Visser, 2008; Gienapp et al., 2008) and society and lead to disproportionate impacts especially on vulnerable communities.

*Code and data availability.*  All model code used in this study is publicly available. The source code and example datasets for the AM2
atmosphere model, the FMS coupler, and the LaD land model are available via the MOM5 Github repository (Griffies et al., 2020). The code for the slab ocean model is also available on Github (GFDL Team, 2019). The simulated data for the analysis in this publication and the code to produce the figures will be made available after publication via zenodo.

*Author contributions.*  AL proposed the idea of the study. AK set up the model configuration, run the simulations and performed the analysis. Both authors analysed the results. AK wrote the paper with contributions by AL.

*Competing interests.*  At least one of the (co-)authors is a member of the editorial board of Earth System Dynamics.

*Acknowledgements.*  The authors gratefully acknowledge the European Regional Development Fund (ERDF), the German Federal Ministry of Education and Research and the Land Brandenburg for supporting this project by providing resources on the high performance computer system at the Potsdam Institute for Climate Impact Research.



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
