# Peer review of "Does the pace of carbon emissions matter in an atmospheric general circulation model?"

_Earth System Dynamics, 2024_

## Referee Comment (RC1)

**Review of "Does the pace of carbon emissions matter in an atmospheric general circulation model?"**

Summary and comments:

In this work, the authors assess whether the mean state and day-to-day variability of global near-surface temperature and precipitation depend on $CO_2$ concentration increase rates using a small ensemble of GFDL-AM2 simulations, specifically focussing on the atmosphere's role and excluding incluences of a reactive and dynamic ocean.

As the authors reason, theory of the governing atmospheric processes indicates a priori that such path-dependence should not exist when slow ocean processes are eliminated artificially. However, I agree with the authors, that it is worth a quantified assessment and support the trial to sort this out, even if this yields the apparent negative finding: the mean state and day-to-day variability of global near-surface temperature and precipitation in a purely atmospheric setting likely do not depend on the $CO_2$ concentration change rate.

The basic idea to compare inter- and intra-scenario differences as a robustness criterion is good. I believe the analysis presented in this manuscript illustrates the argument, however is not sufficient to prove the point (as the authors state themselves). While I appreciate the effort of the analysis and promote the analysis for and publication of negative findings, I feel this analysis could do much better at this, even when restricted to the limited data produced by the authors.

My main suggestions are:

- go beyond one decade and test multiple time ranges up to 30 years. I see little inhibiting this in de-trended space.
- Aggregate the grid-cell level results to different higher levels
- Compute signal-to-noise ratios at all spatial levels.
- Consider analysing existing AMIP-style simulations (data ocean instead of slab ocean) from the CMIP6. There are different scenarios with different CO2 concentration rates and several models with multiple ensemble members.

Overall, in my opinion, this work needs major revisions to serve its purpose.

Line-by-line comments:

Title:         replace "emissions" with "concentration changes". As far as I understand, you only prescribe CO2 concentrations, not emissions. You are not testing the paces of emissions in a GCM with online carbon cycle modeling. Likewise replace emissions in any equivalent occurrence in the text.

Lines 10-11:   this is a good method. In my opinion it additionally needs more quantitative aggregation. Surface area of significantly different variability, magnitude of the difference field, etc.

Lines 37-38:   make clear, whether you refer to temperature or precipitation here.

Lines 48-49:   Why not a 0.5% scenario?

Lines 70-71:   Is this enough for your study and which kind of potential miss-estimation do you expect from such a simple land model?

Line 78:         How did you determine N = 5 as adequate ensemble size for the task? Later on, you try to argue further analysis is inhibited by lacking data. Please indicate shortly why you designed a macroensemble (vs. small perturbation / microensemble), why is a one year IC steps a good choice?

Lines 79-80:    replace with "too far equatorwards on both hemispheres.". Does the sea ice modeling in "slab mode" mean that the behavior you mention for GFDL-AM2 is prescribed or modeled and emergent? And if prescribed, why can you not prescribe the sea ice cover more realistically?

Line: 83:       Annual or decadal mean GMT?

Line 84:        Why don't you subtract the 10 year mean around initialisation in the parent simulation? This would help to exclude variability from the IC. Additionally, for avoiding influences of model GMT drift, a comparison against an unforced reference simulation (ensemble) would make this more robust.

Figure 1:       enumerate the panels please.

                please unify the y-axes of the CO2 concentrations for comparability.

                There is very little year-to-year variability in the annual mean GMT time series compared to the historical record or Earth System Model simulations. Is this because ocean variability and its feedback are excluded or are there additional other players at work?

Line 86:        on "small deviations": Please quantify how small.

Lines 86-87:    Why is it important to have values centered around zero in their distribution in space? As far as I understand you do not consider distribution in space.

Lines 89-91:    With which parameters did you smooth the timeseries? Did you check whether such smoothening gives results close to results of averaging over more years?

Lines 92-94:    Again, please give details on the parameters and other methodical choices here. Any part of variability you do or do not filter out in this step will not be evaluable downstream in your pipeline and can potentially influence your results of variability estimates. So the reader needs to understand, how you determine your trendline and which frequency ranges of temporal variability you are eliminating beyond a linear trend.

Figure 3:       Please enumerate the panels.

        Panel a):       Why not center this line around 2°C? The phenomena you are trying to show would emerge equivalently. And please highlight the global mean change as vertical line.

        Panel b):       Please plot the field showing actual grid cell values. The contour shading you use hides the models resolution and its effects unnecessarily. This is the case for all maps in this work.
                        The caption and colorbar label do not indicate clearly what you are showing. Is it the change in mean state in a 2°C warming world vs. a 0°C warming world and is re-centered around 0 instead of 2°C or something else? If this is it, the presentation with temperature changes centered around 0 instead of 2°C is highly confusing. Please reconsider.

Also I do not understand the centrality of this figure to the paper you seem to suggest with its size and prominence. If this is just a sanity check that the mean state behaves alright for the following comparative analysis, please present it accordingly.

Your plot seems to have deficiencies at lon = 0 (no color shown). Please fix this.

Finally, please give the reader some help on which values differ significantly from a 0°C world in your model. I recommend grid-cell wise significance testing with a correction for false discovery.

Line 112:       Add "on grid cell level" for clarity.

Lines 114-115:       This seems quite normal when only averaging over 10 years and 5 ensemble members (Compare e.g. AMIP simulations spread).

Figure 4:       enumerate panels, please.

At this level of detail, I am not interested in single ensemble member comparison. like in panels b) and c). Please aggregate the information graphically across all possible inter- and intraensemble differences.

Lines 117-118:       Can you really tell this from visual inspection only. It gives the impression yes, but I would like to see more rigorous analysis here. Please not make the reader aggregate your findings, aggregate them for the reader. Use measures of difference field strength, regional, hemispheric, latitude band areal mean values and show whether your estimates (this holds for all your indicators for both mean state and variability as well as temperature and precipitation) emerge from noise (contrast between members of same ensemble).

Lines 120-123:       So why are you using 10yr mean values  instead of longer averaging? Especially after de-trending, what inhibits an analysis of a range differently long averaging windows around a common warming level?

Figure 5:       So compared to Fig. 3, are these maps now re-centered likewise? As far as I understand, no. Please handle Fig. 3 accordingly. I strongly suggest showing 4% together with 1% also there.

I strongly suggest to use another color scheme for precipitation than for temperature to avoid confusion of the reader with reduction vs. increase of temp vs. precip.

Lines 129-130:       Please give the intra-ensemble uncertainty range in mm/d as well.

Lines 133-136:       Please elaborate. What is your analysis worth when the reader is left with speculation that all the signal should probably be noise.

Lines 137-138:       Valuable context, correct comparison in my opinion. However, when the signal is of same order of magnitude as the noise, there can still be a signal. How about a quantification of signal-to-noise ratios at different aggregation levels? Grid cell, region, hemisphere, global? Absolute value means vs. means would help here for an exact argument.

Lines 145-146: So indeed the sea-ice biases have massive influence on the temperature variability. Please argue more precisely why the GFDL-AM2 estimates are still valuable for an assessment of variability changes.

Lines 159-164: I disagree, quantitative statistical analysis can still help to illustrate how likely it is from the available data that differences in mean states and variability between forcing rates are insignificant.

Lines 180-181: Again, please show this graphically for different levels of aggregation.

Lines 194-196: I agree, this analysis is not sufficient to prove the point. While I appreciate the effort of the analysis and promote the analysis for and publication of negative findings, I feel this analysis could do much better at this, even when restricted to the limited data produced by the authors.

Lines 214: So, I suggest you provide the actual information to the reader, what are known biases of GFDL-AM2 concerning mean state changes and intra-decadal variability? And for indicators, where observations do not give a meaningful reference, how does GFDL behavior compare to other atmospheric models?

Lines 216-217: This is trivial, can be omitted in my opinion. However, there are ensembles of AMIP style simulations (data ocean, but otherwise similar) you could try to use as additional source of data to check other models. Multiple models with 3 and more ensemble members for different rates of concentration increase (e.g. amip-hist vs. amip-4xCO2).
https://esgf-node.ipsl.upmc.fr/search/cmip6-ipsl/ search query: amip, tas, day

Lines 221-223: Please also comment on the impact of the choice of land model on near-surface temperature and precipitation variability. Please also reflect on internal variability coming from carbon feedbacks in the Earth System.

---

## Author Comment (AC1)

**Answers to reviewers' comments regarding "Does the pace of carbon emissions matter in an atmospheric general circulation model?"**

We thank the reviewers for their time to evaluate our manuscript and for sharing their perspective. Their comments have substantially improved the manuscript. Major changes include

- clarification of concentration-driven not emission-driven simulations and adaptation of the title and manuscript text accordingly

- strengthening motivation by studies that have become publicly available in the meantime, namely Katzenberger & Levermann (2025) and Hankel (2025)

- adding clarification regarding processing procedure and methodical choices

- expanding period of analysis from 10 to 20 years

- adding further quantitative insights

- providing more insights regarding robustness checks

- aggregation of results to global level

- adding more detailed discussion of model setup and parameter choices

We believe that these adaptations address the points raised and hope that the revised manuscript will now be considered suitable for publication in *ESD*. Please find below our detailed, point-by-point responses.

**Reviewer #1**

This study compares temperature and precipitation changes at 2°C warming in atmosphere model simulations with a slab ocean in which $CO_2$ increases at 1 %/yr and 4%/yr. The authors find no significant differences.
Whereas the title of the paper implied to me that the authors would perhaps compare simulations in an earth system model with different rates of $CO_2$ emissions but the same cumulative emissions, this is not the case, and the authors rely on simulations with a model with specified $CO_2$ concentration which does not simulate the response to emissions at all.

We fully agree with the reviewer and hope that we have made this clear by specifying the atmospheric general circulation model in the title. We also tried to make it very clear in the abstract and introduction, that our study is based on a concentration-driven setup of the GFDL-AM2 model without an interactive carbon cycle. This design allows us to prescribe $CO_2$ concentration pathways precisely and avoids the additional uncertainties associated with carbon-cycle feedbacks. Our study tries to complement other analysis in the field, as for example, the recent study by Hankel (2024), who investigated the role of $CO_2$ ramping rates on the Atlantic Meridional Overturning Circulation, herein omitting carbon cycle feedbacks while keeping oceanic responses. Our study was aiming to investigate if the atmosphere alone can bear memory - an unlikely but not impossible reaction to $CO_2$ forcing.

We understand, that our original title may have suggested an emissions-based framework and thus caused confusion. To address this, we have revised the title to *"Does the pace of carbon forcing matter*

*in an atmospheric general circulation model?"* and clarified throughout the manuscript that our experiments are based on prescribed $CO_2$ concentrations rather than emissions. We thank the reviewer for pointing out the need to make this distinction more explicit.

*Hankel, C.: The effect of CO2 ramping rate on the transient weakening of the Atlantic Meridional Overturning Circulation, Proceedings of the National Academy of Sciences, 122, e2411357 121, 2024.* https://doi.org/10.1073/pnas.2411357121

Moreover the model has no dynamical ocean, and therefore does not realistically simulate the transient climate response to changing atmospheric forcing.

We fully agree with the reviewer that a slab ocean is a strong simplification compared to a fully coupled ocean and, by design, cannot provide a fully realistic representation of the transient climate response. However, this choice is deliberate: the simplified ocean allows us to focus on the role of the atmosphere while avoiding the complexity of full ocean dynamics. Still, the slab ocean does react to changing forcings, as there is a vertical coupling between the atmosphere and the ocean (or sea-ice and the ocean where applicable). Our study should be viewed as a targeted analysis of atmospheric responses under different forcing rates, rather than as a fully realistic Earth system simulation.

Given that the atmosphere-land system responds to changes in forcing generally on a timescale of months or years at most, there is no reason to expect that its response should be different in the different simulations the authors carry out.

We understand the argument to be that the temporal scale of the external forcing ($CO_2$ concentration increase) is slow, while the atmosphere–land system responds on relatively fast timescales. Thus the system will always have enough time to adapt. Indeed, the atmosphere-land coupling is reacting on very short time scales, with the atmosphere alone often responding within days to weeks.

Recent work (Katzenberger & Levermann, 2025), which has been published in the meantime, suggests that the atmosphere may possess a greater memory than previously assumed — as shown in the context of monsoon dynamics. Therefore, we think that the response-time scales may be slower than previously assumed, which would make it possible to observe differences between the simulations. Also dynamical feedbacks or threshold effects would open the possibility that differences between scenarios may emerge.

*Katzenberger, A. and Levermann, A.: Monsoon Hysteresis reveals atmospheric memory, PNAS, Vol. 122 (19), e24180931, 2025. https://doi.org/10.1073/pnas.2418093122.*

To add a historical perspective, the model set-up is similar to that used by Manabe and Wetherald (1975), albeit with a newer version of the GFDL model. Manabe and Wetherald (1975) explain that their model is only able to simulate the equilibrium climate response to changes in $CO_2$. Also, for example, the IPCC First and Second reports explain how an interactive ocean is necessary to simulate transient climate change.

Link to to paper mentioned by reviewer:
https://journals.ametsoc.org/view/journals/atsc/32/1/1520-0469_1975_032_0003_teodtc_2_0_co_2.xml

We thank the reviewer for this insightful historical context. Indeed, our model setup bears similarity to that of Manabe and Wetherald (1975), e.g. idealized topography and no heat transport via ocean currents. As we explain above, there is an ocean-(sea ice-)atmosphere coupling in our setup and the simplified slab ocean does have heat capacity - keeping the key characteristic as relevant for this research question.

We are aware that the current knowledge as e.g. stated in the first and second IPCC reports, assumes that a fully dynamic ocean is necessary to capture differences in the transient climate response because the timescales of the atmosphere are considered to be too fast. This is exactly what we want to test with our setup, in this study on a global level, motivated by findings in Katzenberger & Levermann 2025, as outlined above.

Our aim is to isolate the sensitivity of the atmosphere to different rates of $CO_2$ concentration increase under prescribed forcing. In this context, a simplified setup provides a controlled framework to examine how the pace of forcing can influence atmospheric states, without the added complexity and feedbacks of a dynamic ocean.

While the authors acknowledge that the model they use is not able to realistically simulate the transient response to changing levels of $CO_2$, they argue that the fact that the atmospheric response is not sensitive to the rate of increase in forcing requires confirmation (ln 13). The authors do not cite any references arguing that the atmosphere might exhibit such a sensitivity. Overall I disagree that this requires confirmation.

Many studies have examined how the climate responds to different magnitudes and types of forcing. However, relatively little attention has been given to its sensitivity to the *rate* of forcing change, despite this rate also being highly uncertain in future emissions scenarios. This gap in the literature has also been highlighted by Harger et al. (2025). Given its relevance for global climate, we are convinced that this is a sufficient motivation for this study.

On top, one central motivation for this study is a project in which we find that the atmosphere is able to store information on time-scales longer than previously thought as outlined above (Katzenberger et al. 2025). On a global level, such memory effects are not identified yet which is why we think this requires further analysis. As this project has been published in the meantime and is now citable, we added this part of our motivation in the introduction.

---

## Author Comment (AC2)

**Answers to reviewers' comments regarding "Does the pace of carbon emissions matter in an atmospheric general circulation model?"**

We thank the reviewers for their time to evaluate our manuscript and for sharing their perspective. Their comments have substantially improved the manuscript. Major changes include

- clarification of concentration-driven not emission-driven simulations and adaptation of the title and manuscript text accordingly

- strengthening motivation by studies that have become publicly available in the meantime, namely Katzenberger & Levermann (2025) and Hankel (2025)

- adding clarification regarding processing procedure and methodical choices

- expanding period of analysis from 10 to 20 years

- adding further quantitative insights

- providing more insights regarding robustness checks

- aggregation of results to global level

- adding more detailed discussion of model setup and parameter choices

We believe that these adaptations address the points raised and hope that the revised manuscript will now be considered suitable for publication in *ESD*. Please find below our detailed, point-by-point responses.

**---------------------------**

**Reviewer #2**

**Review of "Does the pace of carbon emissions matter in an atmospheric general circulation model?"**

Summary and comments:

In this work, the authors assess whether the mean state and day-to-day variability of global near- surface temperature and precipitation depend on $CO_2$ concentration increase rates using a small ensemble of GFDL-AM2 simulations, specifically focussing on the atmosphere's role and excluding incluences of a reactive and dynamic ocean.

As the authors reason, theory of the governing atmospheric processes indicates a priori that such path-dependence should not exist when slow ocean processes are eliminated artificially. However, I agree with the authors, that it is worth a quantified assessment and support the trial to sort this out, even if this yields the apparent negative finding: the mean state and day-to-day variability of global near-surface temperature and precipitation in a purely atmospheric setting likely do not depend on the $CO_2$ concentration change rate.

The basic idea to compare inter- and intra-scenario differences as a robustness criterion is good. I believe the analysis presented in this manuscript illustrates the argument, however is not sufficient to prove the point (as the authors state themselves). While I appreciate the effort of the analysis and promote the analysis for and publication of negative findings, I feel this analysis could do much better at this, even when restricted to the limited data produced by the authors.

My main suggestions are:

- go beyond one decade and test multiple time ranges up to 30 years. I see little inhibiting this in de-trended space.

- Aggregate the grid-cell level results to different higher levels

- Compute signal-to-noise ratios at all spatial levels.

- Consider analysing existing AMIP-style simulations (data ocean instead of slab ocean) from the CMIP6. There are different scenarios with different CO2 concentration rates and several models with multiple ensemble members.

Overall, in my opinion, this work needs major revisions to serve its purpose.

We thank Reviewer 2 for the detailed and attentive review and appreciate the encouraging words regarding the value of publishing negative findings—a perspective we fully share. We carefully considered all comments and revised the manuscript accordingly. We believe these changes have strengthened the clarity and robustness of our study.

Line-by-line comments:

Title: replace "emissions" with "concentration changes". As far as I understand, you only prescribe CO2 concentrations, not emissions. You are not testing the paces of emissions in a GCM with online carbon cycle modeling. Likewise replace emissions in any equivalent occurrence in the text.

We thank the reviewer for pointing this out and fully agree that the current title may be misleading. Accordingly, we changed it to *"Does the pace of carbon forcing matter in an atmospheric general circulation model?"*. We also state very clear in the abstract that we work with changing concentration rates. Besides, we clarified any occurrences throughout the text.

Lines 10-11: this is a good method. In my opinion it additionally needs more quantitative aggregation. Surface area of significantly different variability, magnitude of the difference field, etc.

We provide the answer to this comment below in the answer to comment l. 117/118 and adapted the abstract accordingly.

Lines 37-38: make clear, whether you refer to temperature or precipitation here.

The reviewer is right, this information was lacking. The sentence refers to temperature which we clarified in the revised manuscript.

Lines 48-49: Why not a 0.5% scenario?

We thank the referee for the suggestion. In our study, the goal is not to focus on realistic emission scenarios, but to investigate the differences between high and low rates of $CO_2$ increase. The exact choice of rates is therefore not critical; what matters is the relative contrast between slower and faster increases.

Lines 70-71: Is this enough for your study and which kind of potential miss-estimation do you expect from such a simple land model?

We thank the reviewer for this important question. We deliberately chose a very simple land model in order to specifically isolate and examine atmospheric processes. In this way, we can avoid the added complexity of more detailed land-surface interactions. We added a clarification in the manuscript.

Line 78: How did you determine $N = 5$ as adequate ensemble size for the task? Later on, you try to argue further analysis is inhibited by lacking data. Please indicate shortly why you designed a macroensemble (vs. small perturbation / microensemble), why is a one year IC steps a good choice?

We thank the referee for the comment. The ensemble size of $N = 5$ was chosen as a compromise between computational cost and the need to capture variability arising from differing initial conditions.

We designed a macroensemble to capture the variability resulting from substantially different initial conditions, rather than small perturbations around a single state. Using one-year steps between initial conditions allows the ensemble members to sample more independent atmospheric states, as seasonal changes produce substantial differences in each subsequent start year. This ensures that the ensemble spread reflects natural variability rather than potentially correlated perturbations.

Lines 79-80: replace with "too far equatorwards on both hemispheres.". Does the sea ice modeling in "slab mode" mean that the behavior you mention for GFDL-AM2 is prescribed or modeled and emergent? And if prescribed, why can you not prescribe the sea ice cover more realistically?

Replaced as proposed by the reviewer. The sea ice is run in slab mode, meaning it is simulated rather than prescribed. We adapted the wording in the manuscript to make this more clear.

Line: 83: Annual or decadal mean GMT?

We thank the reviewer for pointing out that this might not be clear to the reader in the current formulation and added a clarification that we refer to the annual mean here.

Line 84: Why don't you subtract the 10 year mean around initialisation in the parent simulation? This would help to exclude variability from the IC. Additionally, for avoiding influences of model GMT drift, a comparison against an unforced reference simulation (ensemble) would make this more robust.

We thank the reviewer for this suggestion. In principle, subtracting a 10-year mean around initialization could indeed help reduce variability. However, a 10-year mean around initialization would require the 5 years following initialization. However, these 5 years already contain a trend, so the data is not

stationary. Therefore, while this approach could theoretically reduce variability, it is not feasible in practice.

We also thank the reviewer for the suggestion regarding an unforced ensemble. Excluding a potential GMT drift is indeed a valid approach to make the findings more robust. In fact, we already have such an unforced simulation: the 20-year equilibration period under constant $CO_2$ effectively serves as a reference run. During this period, no noticeable drift in global mean temperature is observed. Therefore, the differences seen in the $CO_2$ increase simulations can be confidently attributed to the varying emission rates.

It is also interesting to note that even if small drifts were present, they would affect all simulations equally. Since we focus on relative differences, such drifts would not influence our results.

Figure 1: enumerate the panels please.

Done.

please unify the y-axes of the CO2 concentrations for comparability.

Done.

There is very little year-to-year variability in the annual mean GMT time series compared to the historical record or Earth System Model simulations. Is this because ocean variability and its feedback are excluded or are there additional other players at work?

We thank the reviewer for this interesting observation. Compared to observations (e.g., https://berkeleyearth.org/global-temperature-report-for-2023/), the annual mean GMT variability in our simulations is slightly smaller, which is indeed expected due to the limited ocean depth, the ocean's heat capacity, and the absence of ocean currents in our simplified setup. However, the difference is not as large as it may appear. Our Figure emphasizes daily variability, which strongly exceeds the annual variability, giving the visual impression that the annual variability is comparatively small.

Line 86: on "small deviations": Please quantify how small.

We added a quantification at the respective sentence.

Lines 86-87: Why is it important to have values centered around zero in their distribution in space? As far as I understand you do not consider distribution in space.

Centering values around zero is a normalization step that allows us to more clearly identify regions of relatively stronger or weaker warming, independent of the absolute global mean increase. While we do not explicitly analyze the full spatial distribution statistically, this approach facilitates visual comparison and highlights spatial patterns, as illustrated in Fig. 3.

Lines 89-91: With which parameters did you smooth the timeseries? Did you check whether such smoothening gives results close to results of averaging over more years?

As explained in the respective lines and illustrated in Fig. 2, step 2, we first average over all 20 years. For the result, we apply singular spectrum analysis (SSA) to achieve further smoothening. The applied parameter in the SSA is now specified inthe manuscript. We also checked different window sizes and

the results are robust with regard to varying window sizes. We added the key parameter of the singular spectrum analysis in the text to clarify the details.

Lines 92-94: Again, please give details on the parameters and other methodical choices here. Any part of variability you do or do not filter out in this step will not be evaluable downstream in your pipeline and can potentially influence your results of variability estimates. So the reader needs to understand, how you determine your trendline and which frequency ranges of temporal variability you are eliminating beyond a linear trend.

We agree that providing details on this step is important and aim for a concise, on-point explanation. The trend of the deseasonalized time series is determined by calculating annual means and smoothing them using singular spectrum analysis (SSA). We then subtract this first-order trend and use the resulting 10-year window as the basis for our analysis. No additional filtering is applied, so all higher-frequency variability is retained, ensuring that the downstream variability estimates reflect the natural fluctuations beyond the removed first-order trend. We added clarification regarding the key parameters.

Figure 3: Please enumerate the panels.

Done.

Panel a): Why not center this line around 2°C? The phenomena you are trying to show would emerge equivalently. And please highlight the global mean change as vertical line.

We added a vertical line at the GMT change of 2°C.

Panel b): Please plot the field showing actual grid cell values. The contour shading you use hides the models resolution and its effects unnecessarily. This is the case for all maps in this work.

We thank the reviewer for the suggestion. While plotting the raw grid-cell values would indeed show the model resolution more explicitly, we chose contour shading to provide a clearer visual representation of the spatial patterns and regional differences. The smoothing does not affect the qualitative interpretation of the results, as the underlying grid-cell variability is preserved in the analysis. Therefore, we believe that the current presentation effectively communicates the main findings without compromising scientific accuracy.

The caption and colorbar label do not indicate clearly what you are showing. Is it the change in mean state in a 2°C warming world vs. a 0°C warming world and is re-centered around 0 instead of 2°C or something else? If this is it, the presentation with temperature changes centered around 0 instead of 2°C is highly confusing. Please reconsider.

We thank the reviewer for pointing out that the presentation might be confusing. We spatially normalize the data for the 10-years period around the timestep with an 2°C GMT increase, as explained in more detail above. We adapted the text and caption in order to remove any potential confusion.

Also I do not understand the centrality of this figure to the paper you seem to suggest with its size and prominence. If this is just a sanity check that the mean state behaves alright for the following comparative analysis, please present it accordingly.

This figure (assumed to be Fig. 3) is included to provide insight into how the mean state of temperature at 2°C global mean warming appears in detail, illustrating the baseline state that we analyze throughout

the paper. It is intended as contextual information for the comparative analysis of variability, rather than as a central result. The figure size was chosen to make the details clearly visible. As usual, the final presentation will be determined by the journal during typesetting.

Your plot seems to have deficiencies at lon = 0 (no color shown). Please fix this.

This is a result of the projection.

Finally, please give the reader some help on which values differ significantly from a 0°C world in your model. I recommend grid-cell wise significance testing with a correction for false discovery.

We thank the reviewer for this suggestion. While grid-cell wise significance testing with correction for multiple comparisons is useful when assessing absolute deviations from a reference state, our study focuses on relative differences between $CO_2$ increase rates and their impact on variability, rather than on absolute changes from a 0 °C baseline. Given our ensemble size and the comparative nature of the analysis, we believe that the current presentation adequately conveys the main findings. If the changes are significantly different from 0°C is not relevant for our research question.

Line 112: Add "on grid cell level" for clarity.

Done, thanks for the suggestion that indeed improves clarity.

Lines 114-115: This seems quite normal when only averaging over 10 years and 5 ensemble members (Compare e.g. AMIP simulations spread).

We agree with the reviewer.

Figure 4: enumerate panels, please.

Not necessary anymore, as we removed the panels, see below.

At this level of detail, I am not interested in single ensemble member comparison. like in panels b) and c). Please aggregate the information graphically across all possible inter- and intraensemble differences.

We thank the reviewer for this comment and removed panel b and c as proposed to avoid unnecessary detail. Panel a shows the aggregated information, namely the difference between ensemble means.

Lines 117-118: Can you really tell this from visual inspection only. It gives the impression yes, but I would like to see more rigorous analysis here. Please not make the reader aggregate your findings, aggregate them for the reader. Use measures of difference field strength, regional, hemispheric, latitude band areal mean values and show whether your estimates (this holds for all your indicators for both mean state and variability as well as temperature and precipitation) emerge from noise (contrast between members of same ensemble).

We thank the reviewer for this comment and added aggregated results in the revised manuscript.

Lines 120-123: So why are you using 10yr mean values instead of longer averaging? Especially after de-trending, what inhibits an analysis of a range differently long averaging windows around a common warming level?

We thank the reviewer for this valuable observation. Indeed, using a longer window is also possible after detrending. We changed to a window size of 20 years with the results remaining robust towards these changes.

Figure 5: So compared to Fig. 3, are these maps now re-centered likewise? As far as I understand, no. Please handle Fig. 3 accordingly. I strongly suggest showing 4% together with 1% also there.

We thank the reviewer for this attentive comment. Indeed, the maps in Fig. 5 are not re-centered as in Fig. 3. The reason is that we are comparing the states within the time window around the 2 °C GMT threshold. For temperature, this allows centering around the defined warming level, but for precipitation there is no analogous threshold value. Therefore, re-centering in the same way is not possible.

I strongly suggest to use another color scheme for precipitation than for temperature to avoid confusion of the reader with reduction vs. increase of temp vs. precip.

We thank the reviewer for this suggestion. Precipitation and temperature represent different variables. Both are displayed here as anomalies, where negative values are shown in blue and positive values in red. This red–blue color scheme is widely used in climate science for anomaly fields and provides a clear and consistent way to distinguish decreases from increases across variables. We therefore prefer to retain the current scheme.

Lines 129-130: Please give the intra-ensemble uncertainty range in mm/d as well.

Done.

Lines 133-136: Please elaborate. What is your analysis worth when the reader is left with speculation that all the signal should probably be noise.

We thank the reviewer for the comment and added further elaboration in the respective paragraph.

Lines 137-138: Valuable context, correct comparison in my opinion. However, when the signal is of same order of magnitude as the noise, there can still be a signal. How about a quantification of signal-to-noise ratios at different aggregation levels? Grid cell, region, hemisphere, global? Absolute value means vs. means would help here for an exact argument.

It is great to hear that the reviewer agrees with our assessment. We added a quantification to make the argument more specific.

Lines 145-146: So indeed the sea-ice biases have massive influence on the temperature variability. Please argue more precisely why the GFDL-AM2 estimates are still valuable for an assessment of variability changes.

Indeed, sea-ice biases strongly affect the absolute magnitude of temperature variability. However, the GFDL-AM2 estimates remain valuable for our assessment because we focus primarily on relative changes in variability across experiments rather than the exact baseline values. Figure S10 illustrates that key spatial features of variability—such as enhanced variability over land and the poleward increase—are consistently reproduced across all simulations (including the 1% and 4% experiments). This robustness ensures that the relative comparisons of variability changes are not compromised by the sea-ice bias. In addition, we provide global spatial maps, which further minimize the influence of regional biases on our overall conclusions.

Lines 159-164: I disagree, quantitative statistical analysis can still help to illustrate how likely it is from the available data that differences in mean states and variability between forcing rates are insignificant.

We thank the reviewer for this remark. We have added further aggregation as proposed in other comments and adapted the corresponding sentence. However, given the limited ensemble size, the power of quantitative statistical tests remains limited.

Lines 180-181: Again, please show this graphically for different levels of aggregation.

Done.

Lines 194-196: I agree, this analysis is not sufficient to prove the point. While I appreciate the effort of the analysis and promote the analysis for and publication of negative findings, I feel this analysis could do much better at this, even when restricted to the limited data produced by the authors.

We thank the reviewer for this comment and think that it is resolved thanks to the quantifications we added in the context of this review. We adapted the respective sentence accordingly.

Lines 214: So, I suggest you provide the actual information to the reader, what are known biases of GFDL-AM2 concerning mean state changes and intra-decadal variability? And for indicators, where observations do not give a meaningful reference, how does GFDL behavior compare to other atmospheric models?

We added further details regarding the limitations of GFDL-AM2.

Lines 216-217: This is trivial, can be omitted in my opinion. However, there are ensembles of AMIP style simulations (data ocean, but otherwise similar) you could try to use as additional source of data to check other models. Multiple models with 3 and more ensemble members for different rates of concentration increase (e.g. amip-hist vs. amip-4xCO2). https://esgf-node.ipsl.upmc.fr/search/cmip6-ipsl/ search query: amip, tas, day

We thank the reviewer for this suggestion. We have omitted the respective sentence as proposed. We also appreciate the idea to compare our results with AMIP-style simulations. To our knowledge, there are simulations with 1% increase, but not with other rates of increase which would be necessary for the research question of our study. The 4xCO2 simulation is of different nature, as the concentration levels are set abruptly to this new level.

Lines 221-223: Please also comment on the impact of the choice of land model on near- surface temperature and precipitation variability. Please also reflect on internal variability coming from carbon feedbacks in the Earth System.

We thank the reviewer for raising these important points. The choice of land model can indeed affect near-surface temperature and precipitation variability e.g. through processes such as soil moisture feedbacks, vegetation response, run-off timing or albedo effects. We deliberately chose a simplified land module to exclude the uncertainties entering due to these dynamics.

Similarly, internal variability associated with carbon cycle feedbacks could introduce additional fluctuations in $CO_2$ concentrations and hence amplify or dampen variability in temperature and precipitation. In our concentration-driven simulations, $CO_2$ pathways are prescribed, so variability arises solely from atmospheric dynamics. Including an interactive carbon cycle would move the setup

closer to reality, but at the same time broaden the focus beyond the dynamics considered in the present study.

We added these aspects to the discussions.